# Remote Sensing Image Semantic Segmentation Based on Edge Information Guidance

**Chu He** [1,2,*] ⬢**, Shenglin Li** [1]**, Dehui Xiong** [1]**, Peizhang Fang** [1] **and Mingsheng Liao** [2]

[1] Electronic Information School, Wuhan University, Wuhan 430072, China; lsl2015@whu.edu.cn (S.L.); dhuixiong@whu.edu.cn (D.X.); fpz@whu.edu.cn (P.F.)

[2] State Key Laboratory for Information Engineering in Surveying, Mapping and Remote Sensing, Wuhan University, Wuhan 430079, China; liao@whu.edu.cn

[*] Correspondence: chuhe@whu.edu.cn; Tel.: +86-133-0716-2028

**Abstract:** Semantic segmentation is an important field for automatic processing of remote sensing image data. Existing algorithms based on Convolution Neural Network (CNN) have made rapid progress, especially the Fully Convolution Network (FCN). However, problems still exist when directly inputting remote sensing images to FCN because the segmentation result of FCN is not fine enough, and it lacks guidance for prior knowledge. To obtain more accurate segmentation results, this paper introduces edge information as prior knowledge into FCN to revise the segmentation results. Specifically, the Edge-FCN network is proposed in this paper, which uses the edge information detected by Holistically Nested Edge Detection (HED) network to correct the FCN segmentation results. The experiment results on ESAR dataset and GID dataset demonstrate the validity of Edge-FCN.

**Keywords:** remote sensing image; semantic segmentation; edge information; Edge-FCN

## 1. Introduction

### 1.1. Background

Remote sensing images may include a variety of geomorphological information, such as roads, arable land, and buildings. To classify this different geomorphological information is of great significance for topographic surveys and military analysis. To finish the classification task, each pixel in a remote sensing image should be assigned to a label associated with a terrain category, which is consistent with image semantic segmentation.

Image semantic segmentation plays a critical role in computer vision, the task of which is to assign a semantic label to each pixel in an image. Traditional algorithms for image semantic segmentation generally consist of a feature extractor and a classifier, such as the work in [1]. Although the traditional algorithm is efficient enough, it cannot meet the needs of high accuracy. With the successful application of Convolution Neural Network (CNN) [2] in the field of computer vision, researchers have begun to consider using CNN in semantic segmentation [3]. A lot of creative algorithms such as FCN [4], Deeplabs [5–7], CRF as RNN [8] etc., have made surprising results. Fully Convolution Network (FCN) is the first end-to-end network for semantic segmentation, which creatively introduces deconvolution. Deeplabs mainly relies on Dilated Convolution and post-processing Conditional Random Forest (CRF) [9] to refine the segmentation results. CRF as RNN [8] makes the CRF integrate into the segmentation network to form an end-to-end network.

Since CNN has made an impressive achievement on image semantic segmentation, let us quickly review the excellent algorithms proposed recently. Generally, these networks can be divided into the following five aspects.

**Encoder–Decoder:** FCN [4] is the first network for semantic segmentation with an encoder–decoder structure. The idea of SegNet [10,11] is very similar to FCN, whereas it encodes and decodes each size of feature map and uses max-pooling indices for upsampling. U-net [12] introduces very low-level information that is effective for recovering details. Consequently, it is usually used for medical image segmentation. Fine Segmentation Network (FSN) proposed in [13] also follows the encoder–decoder paradigm in semantic labeling of high-resolution aerial imagery and LiDAR data.

**Atrous Convolution Base:** Atrous convolution is proposed for enlarging the reception field by inserting "0" in filters without increasing the computation. It is widely used in Deeplab-V1 [5], Deeplab-V2 [6], and Deeplab-V3 [7]. To improve the density of output class maps, ref. [14] introduces atrous convolution in high-resolution remote sensing image classification. Moreover, Atrous Spatial Pyramid Pooling (ASPP) implemented in Deeplab-V2 [6] is also used in remote sensing image classification [15].

**Multi-Level Fusion:** It is well known that there is more spatial information in low-level feature maps while the high-level feature maps are richer in semantic information. Semantic segmentation is a joint task of localization and classification requiring both spatial information and semantic information. As a result, multi-level fusion is widely used in recent networks, such as RefineNet [16], PSPNet [17], and GCN [18]. Algorithms in [19] employ CNN to generate five-level features and then a linear model is used to fuse the features of different levels. A hierarchical multi-scale CNN with auxiliary classifiers is proposed in [20] to learn hierarchical multi-scale spectral–spatial features for HSI classification.

**CNN Integrate Traditional Algorithm:** It is also popular in combining CNN and traditional algorithms in remote sensing image segmentation. Some researchers integrate a graph embedding model and FCN [21] to extract both the shallow-linear and deep-nonlinear features to segment the remote sensing image more accurately. Methods in [14,22] both further refine the output class maps generated from CNN using Conditional Random Field (CRF) post-processing. Texture analysis [23] is also widely used in semantic segmentation of remote sensing images [24], such as classification of land cover of a Mediterranean region [25] and road traffic condition classification [26].

**Boundary Refinement:** Boundary detection is also a fundamental challenge regarding image understanding. Lots of specific methods for detecting boundaries have been proposed recently in [27–29]. What they have in common is that they straightly concatenate the different level of features to extract the boundary. Discriminative Feature Network (DFN) [30] is proposed for tacking the intra-class inconsistency problem and inter-class indistinction problem. In contrast to DFN, our model constrains better segmentation by finding changes in the image signal and the nature of the region signal.

Some approaches have been proposed recently on introducing edges into the semantic segmentation network. A method for correcting segmentation results using edges is proposed in [31]. It draws on the Domain Transform method of one-dimensional signal, and uses the edge intensity as the weight of the filter to correct the original segmentation result. The diffusion method is improved in [32], and the edge distance map is proposed to guide the direction of diffusion. Both methods in [31,32] belong to the method of adding edge information to correct the segmentation after the segmentation is completed. What we have done in this paper is to use an end-to-end network to combine semantic segmentation with edge detection so that the associated parameters can be updated by training.

*1.2. Problem and Motivation*

FCN is considered to be the landmark network since it uses the end-to-end network for the first time in semantic segmentation and has achieved satisfying results. However, many problems still exist in FCN: First, the result obtained by upsampling is still rough so some detailed information in the image may not be acquired. Secondly , the relevance between pixels is not fully used and thus the spatial consistency is lost. Thirdly, it lacks a priori knowledge constraints. To segment the remote sensing image more accurately, some researchers focus on integrating a graph embedding model and FCN to extract both the shallow-linear and deep-nonlinear features [21]. Although these methods can

significantly improve the segmentation results, the priori information is still not considered. Motivated by the successful application of prior knowledge in remote sensing image scene classification [33], we naturally consider adding prior knowledge to remote sensing image semantic segmentation.

Edge detection is used to find out the obvious changes in brightness in the image. Therefore, it can eliminate irrelevant information in the image and preserve important structural properties. Traditional edge detection operators are similarly based on the gradient of the image signal to extract the edge information, such as Sobel operator, Roberts operator, and Canny operator [34], etc. Although using traditional operators can extract edge information fast, it may fail to get the edge information between different categories, which is needed for semantic segmentation. Holistically Nested Edge Detection (HED) network [35] is a deep network designed for edge detection. It produces multi-scale feature maps and multiple loss functions to perform backpropagation, which can provide the edge information we want.

### 1.3. Structure and Contribution

To tackle the problems in FCN, one way is to combine the FCN and HED to correct the FCN segmentation result by the possibility of each pixel as an edge point detected by HED. Accordingly, a new segmentation result can be acquired.

From the perspective of signal processing, the segmentation network is like a low-pass filter, which can smooth the image signal and assign the pixels in the similar region with the same semantic label. On the contrary, the edge detection network is like a high-pass filter, which can amplify the distinction of features and extract the semantic boundaries. By combining FCN and HED, the edge scores produced by HED can constrain the segmentation results and thus renew them.

The newly proposed network is named Edge-FCN, seen in Figure 1. According to different ways of combination, they are respectively called Cascade-Edge-FCN and Correct-Edge-FCN.

**Cascade-Edge-FCN** It directly concatenates the segmentation score map produced by FCN and the edge score map produced by HED and then recovers the new edge score map and new segmentation score map by using convolution layers.

**Correct-Edge-FCN** Based on the idea that the larger the edge score, the larger the correction should be, Correct-Edge-FCN uses edge score map to correct segmentation result, which can be realized by a convolution layer.

In summary, the main contributions of this paper are as follows:

(1) Edge information is used as the a priori knowledge to guide remote sensing image segmentation.
(2) Two conceptually simple end-to-end networks are proposed in this paper by combining FCN and HED, which can be trained and inferenced easily without complicated procedures.
(3) Learning from the point in HED, multiple loss fusion is applied to Edge-FCN. Therefore, deep supervision can be realized for each layer when training.

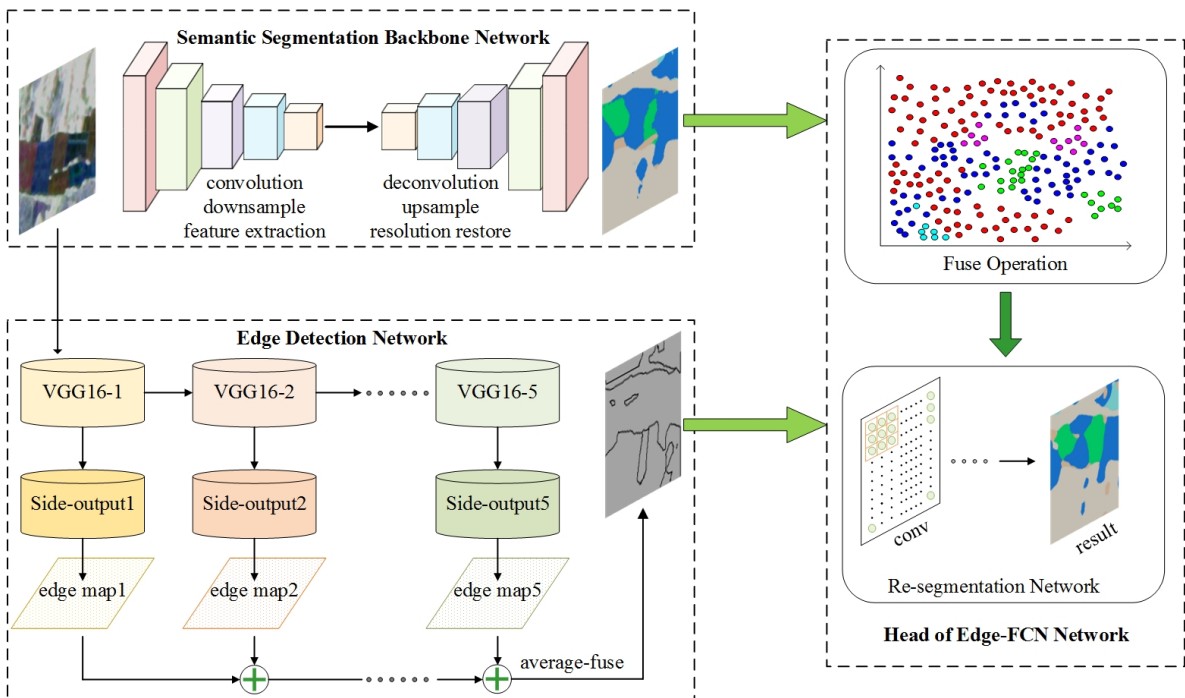

**Figure 1.** Framework of proposed method Edge-FCN. The semantic segmentation result and edge detection result are both input into the Fusion Operation, and the edge-guided segmentation result can be obtained through the Re-segmentation Network.

## 2. Preliminaries

### 2.1. FCN Framework

Fully convolution Network (FCN) is a powerful network for image semantic segmentation, which is widely used for dense prediction. FCN is fine-tuned from the pretrained VGG-16 [36] network, which can achieve end-to-end and pixel-to-pixel prediction. VGG-16 network is mainly composed of 13 convolution layers (each convolution layer is followed by a RELU layer), 5 pooling layers for downsampling, 3 fully connected layers and a SoftMax [37,38] layer for classification. The FCN structure is based on VGG-16, replacing 3 fully connected layers with 2 convolution layers. By combining feature maps of different resolutions, there are three FCN architectures—single-flow FCN-32s, dual-tributary FCN-16s, and three-tributary FCN-8s—as shown in Figure 2. After lots of contrast experiments and analysis, the designer of FCN proves that FCN-8s performs best in the semantic segmentation task.

VGG-16 is originally designed for the whole image classification task. Therefore, the following modifications are necessary to convert it into a semantic segmentation network. First, all fully connected layers are substituted by convolutional layers to ensure the nets can take arbitrary-sized inputs and produce 2D spatial outputs, because the fully connected layers just have fixed dimensions and generate a feature vector. Secondly, to refine the spatial information, a skip architecture is developed to combine fine layers and coarse layers. Thirdly, to output the same size as the input image, the fully convolutional layers are followed by several deconvolution layers, since the output spatial dimensions of ahead layers are reduced by subsampling.

What is commonly used for upsampling is interpolation. For instance, simple bilinear interpolation computes each output $y_{ij}$ from the nearest four inputs by a linear map that depends only on the relative positions of the input and output cells. Obviously, parameters in simple bilinear interpolation algorithm can never be changed once they are determined. For deconvolution in FCN, it realizes upsampling like this: First, it inserts $f-1$ "0" around each input cells. Here, $f$ denotes the upsampling factor. Secondly, a convolution operation is used for the processed input data. It can be

seen that deconvolution is essentially the same as convolution. The upsampling factor $f$ is equal to stride $s$ of the convolution layer. One concrete deconvolution operation can be seen in Figure 3.

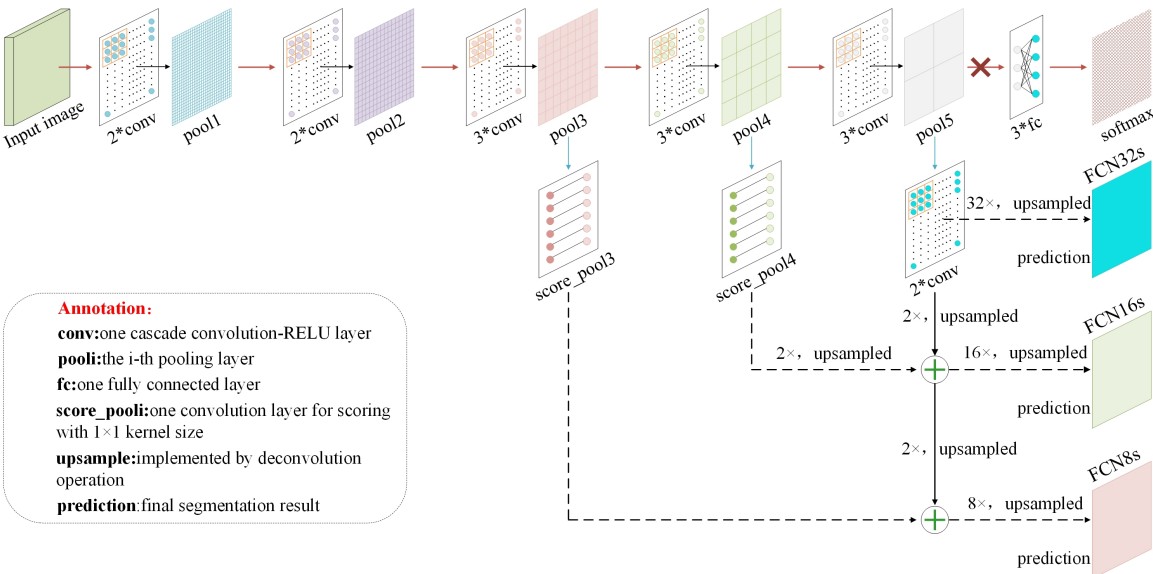

**Figure 2.** Architecture of Fully Convolution Network adapted from VGG-16. It can be seen that FCN8s incorporates more levels of feature maps by skip-connection. Therefore, its segmentation result is more refined and used as the segmentation backbone network in this paper.

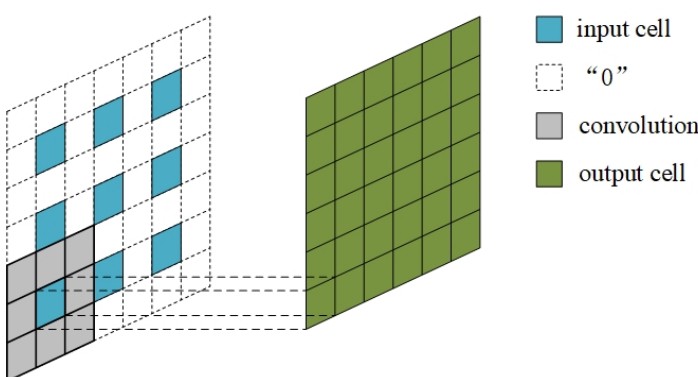

**Figure 3.** Deconvolution operation with upsampling factor $f = 2$ and $stride = 1$. The shape of the input data in the figure is $3 \times 3$, and the shape of the output data through deconvolution becomes $6 \times 6$, which achieves $2\times$ upsampling.

### 2.2. HED Framework

Holistically Nested Edge Detection (HED) network is an end-to-end edge detection network which can realize deep-supervision and multi-scale fusing. The model can train and predict edges in an image-to-image fashion because of its holistic structure. With nested strategy, it emphasizes the inherited and progressively refined edge maps produced as side outputs.

HED is adapted from VGG-16 by making the following modifications: (a) Connecting the side-output layer to the last convolution layer in each convolution stage, respectively conv1_2, conv2_2, conv3_3, conv4_3, conv5_3. (b) Cutting the last stage of VGGNet, including the 5th pooling layer and all the fully connected layers. The side-output layer mentioned above generates the inherent scales of edge maps, consisting of one convolution layer with kernel size of $1 \times 1$ and one deconvolution layer for upsampling. Each of the edge map forms a loss function with the ground truth. Finally, all the edge maps are fused to a weighted edge map prediction. The weighted edge score map prediction

also forms a loss function with the ground truth. By fusing these loss functions, the HED network can realize deep-supervision and multi-scale training and prediction.

Architecture of HED network can be seen in Figure 4. It can be seen that the five blocks of HED network all produce an edge map. Obviously, the front edge maps have richer details while the back-edge maps have richer semantic information. Therefore, the final fused edge map has both rich semantic information and spatial details. Moreover, each edge map generates a loss function with the edge ground truth, which is conducive to achieving strong supervision of each layer of the network and enhancing the learning ability of the whole network.

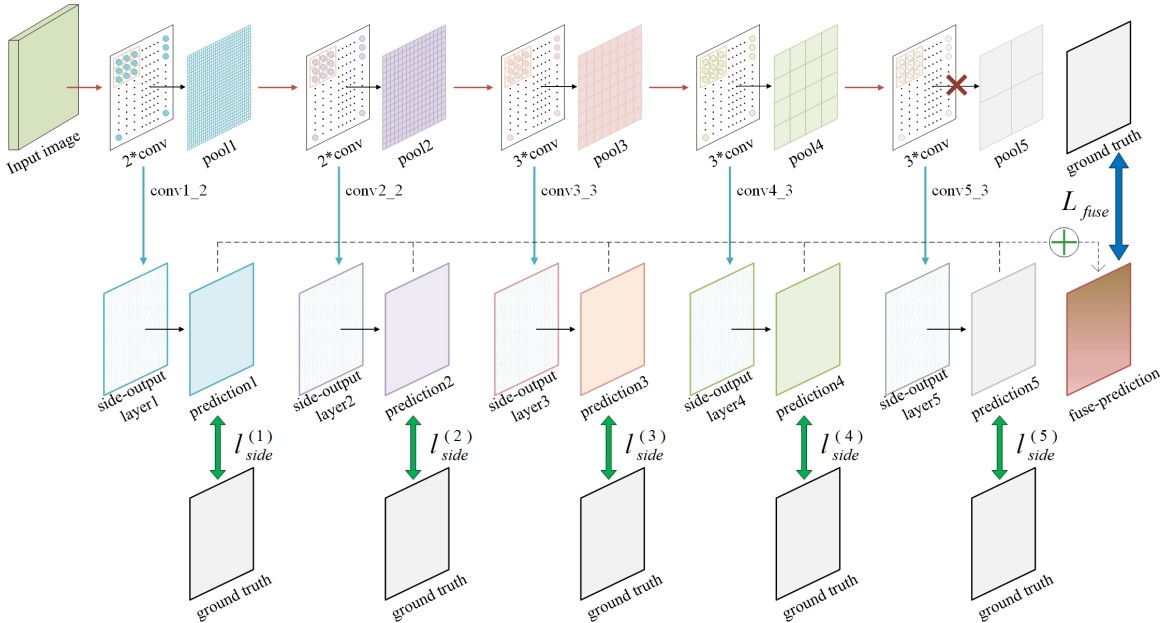

**Figure 4.** Architecture of HED network. The side-output layer here is composed a convolution layer with kernel size of $1 \times 1$ and a deconvolution layer for upsampling. All loss functions are fused.

Calculation details of HED are as follows.

We denote the input training dataset by $T = \{(X_n, Y_n)|n = 1, 2, ..., N\}$, where $X_n = \{x_j^{(n)}, j = 1, 2, ..., |X_n|\}$ is the input image and $Y_n = \{y_j^{(n)}, j = 1, 2, ..., N|y_j^{(n)} \in \{0, 1\}\}$ is the corresponding ground truth binary edge map. For simplicity, all network layer parameters are denoted as $W$. There are 5 side-output layers in HED, in which the corresponding weights are denoted as $w = (w^{(1)}, ..., w^{(5)})$. Each side-output has the class-balanced cross-entropy loss function

$$l_{side}^{(m)}(W, w^{(m)}) = -\beta \sum_{j \in \{Y_+\}} logPr(y_j = 1|X; W, w^{(m)}) - (1 - \beta) \sum_{j \in \{Y_-\}} logPr(y_j = 0|X; W, w^{(m)}) \quad (1)$$

where $\beta = |Y_-|/|Y_+|$ denotes the ratio of non-edge labels and edge labels in ground label sets. $Pr(y_j = 1|X; W, w^{(m)}) = \sigma(a_j^{(m)})$ is computed using sigmoid function $\sigma(.)$ on the activation value at pixel $j$. At each side-output layer, each edge map predictions $\hat{Y}_{side}^{(m)} = \sigma(\hat{A}_{side}^{(m)})$ can be obtained, where $\hat{A}_{side}^{(m)} = \{a_j^{(m)}, j = 1, ..., |Y|\}$ are activations of layer $m$.

To directly use side-output predictions, a "weighted-fusion" operation is used here. That is $\hat{Y}_{fuse} = \sigma(\sum_{m=1}^{5} h_m \hat{A}_{side}^{(m)})$, where $h_m$ is the fusion weight. The fusion output $\hat{Y}_{fuse}$ also produces the class-balanced cross-entropy loss function $L_{fuse}(W, w, h)$. Therefore, the total loss function $L_{HED}$ of HED network in training phase is

$$L_{HED} = \sum_{m=1}^{5} l_{side}^{(m)}(W, w^{(m)})/5 + L_{fuse}(W, w, h) \quad (2)$$

In inference phase, the final unified output $\hat{Y}_{HED}$ is given by

$$\hat{Y}_{HED} = Average(\hat{Y}_{fuse}, \hat{Y}_{side}^{(1)}, ..., \hat{Y}_{side}^{(5)}) \tag{3}$$

Here, the ground truth of edges is obtained from the ground truth of semantic segmentation. The conversion rule is as follows: When one of the adjacent four pixels of a pixel is different from it, it is regarded as the edge point and labeled as 1; otherwise it is the non-edge point and labeled as 0. Therefore, the edge here is more like the boundary of different categories. The ground truth of the edge is equivalent to a sparse matrix with the same size as the image. It is roughly evaluated that the edge points account for about 2% in a ground truth image.

## 3. Our Work

### 3.1. Annotations

Let the batch size of images be $n$, the number of segmented classes be $c$, the height and width of the image be $h$ and $w$, the output of the FCN network be $S_1$. $S_1$ is a tensor of shape $(n, c, h, w)$, representing the score value of each pixel in the image on each class. The output of the HED network is $E_1$, a tensor of shape $(n, 1, h, w)$, indicating the probability value of each pixel being an edge.

Let the ground truth of the semantic segmentation be $GT\_seg$, and the ground truth of edge detection be $GT\_edge$. Ground truth here is used for computing loss function.

### 3.2. Cascade-Edge-FCN

The structure of the Cascade-Edge-FCN network can be seen in Figure 5.

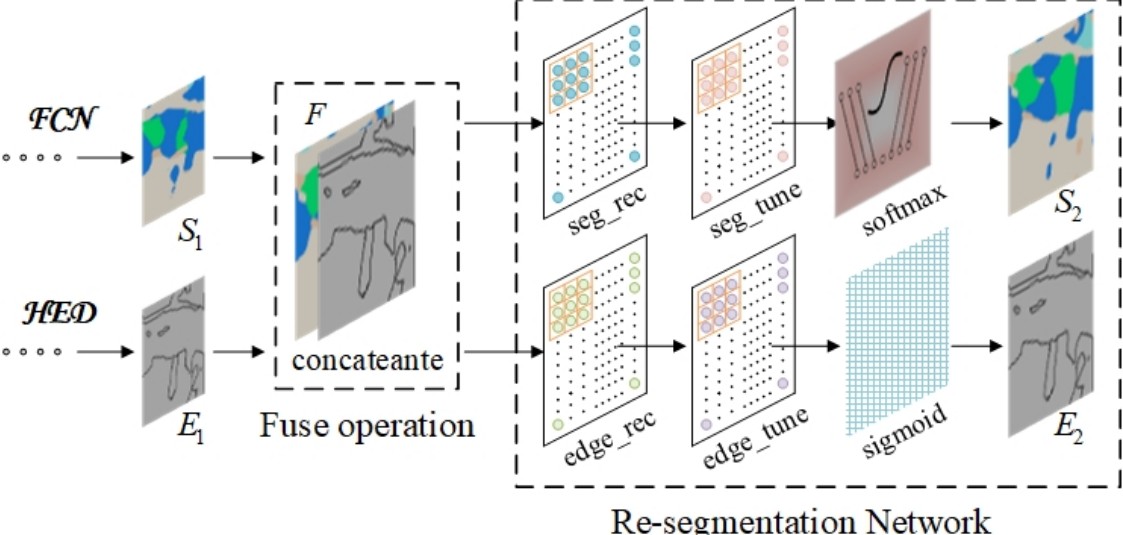

**Figure 5.** Architecture of Cascade-Edge-FCN network. This network connects the original segmentation map with the edge map, and then restores the segmentation map with edge information through the Re-segmentation network.

We cascade $S_1$ and $E_1$ in the dimension of the channel to obtain a tensor $F$ with the channel number $c + 1$. Let the tensor $F$ pass through the convolution layer *seg_rec* used for recovering the segmentation score map's channel, the fine-tuning convolution layer *seg_tune*, and the SoftMax layer to obtain the final score map $S_2$ . That is:

$$S_2 = softmax(seg\_tune(seg\_rec(F))) \tag{4}$$

Similarly, let the tensor $F$ pass through the convolution layer $edge\_rec$ used for recovering the edge score map's channel, the fine-tuning convolution $edge\_tune$ layer, and the sigmoid layer to obtain the final score map $E_2$. That is:

$$E_2 = sigmoid(edge\_tune(edge\_rec(F))) \tag{5}$$

Some parameters of the four convolution layers can be seen in Table 1.

**Table 1.** Parameters of the four convolution layers.

| Name | In-Channels | Out-Channels | Kernel | Stirde | Padding |
|---|---|---|---|---|---|
| seg_rec | $c+1$ | $c$ | $3 \times 3$ | 1 | 1 |
| seg_tune | $c$ | $c$ | $1 \times 1$ | 1 | 0 |
| edge_rec | $c+1$ | 1 | $3 \times 3$ | 1 | 1 |
| seg_rec | 1 | 1 | $1 \times 1$ | 1 | 0 |

Learning from the strategy in HED network, multi-level loss function fusion is applied to Cascade-Edge-FCN. The loss function of the segmentation part adopts the multiple cross-entropy function denoted as $f_1$, and the edge detection part adopts the weighted binary cross-entropy function denoted as $f_2$. $L_1$, $L_2$ and $L$ respectively represent loss function of segmentation part, loss function of edge detection part and total loss function of Cascade-Edge-FCN. We have:

$$L_1 = 0.5f_1(S_1, GT\_seg) + 0.5f_1(S_2, GT\_seg) \tag{6}$$

$$L_2 = 0.5L_{HED} + 0.5f_2(E_2, GT\_edge) \tag{7}$$

$$L = L_1 + L_2 \tag{8}$$

The Casade-Edge-FCN directly cascades the segmentation result generated by the FCN with the edge information generated by the HED, and then obtains a new segmentation result guided by the edge information through the convolution operation. The idea of cascading segmentation information and edge information is similar to the idea of combining low-level features and high-level features in Feature Pyramid Network (FPN) [39]. This proposed network is simple in concept and easy to implement. How to use the edge information to guide the segmentation results is completely learned by the parameters of the added convolution layer.

*3.3. Correct-Edge-FCN*

The Domain Transform in a 1-D signal [31] is as follows: For the original 1-D signal $x$ of length $N$, the output signal $y$ after the Domain Transform is:

$$y_i = \begin{cases} x_i & i = 1 \\ (1 - w_i)x_i + w_i y_{i-1} & i > 1 \end{cases} \tag{9}$$

Here $w$ is the filter coefficient, which is determined by the smoothing coefficient.

According to this formula, suppose that the greater the possibility that the pixel is the edge point is, the larger correction should be done to the original segmentation result. Let the correction value be $Q$, which we get here is the integration of the eight points around the original pixel and its own weighting value. Here a convolution layer can be used to achieve the weighted fusion, so that the fusion parameter of the correction value can be trained. Let the weighted-fusion convolution layer be $seg\_fuse$, whose input channel, output channel, kernel size, stride and padding are respectively set as $c, c, 3 \times 3, 1$ and 1. Let the semantic segmentation score after the guidance of edge score $E_1$ and correction value be $Q$. $Q$ and $S_2$ can be calculated as follows:

$$Q = seg\_fuse(S_1) \tag{10}$$

$$S_2 = (1 - E_1)S_1 + E_1 Q \tag{11}$$

Comparing Equations (9) and (11), three modifications are done here: the weights of the filter $w$ is replaced by $E_1$; the original signal $x$ is replaced by $S_1$; $y_{i-1}$, which only represents one direction, is replaced by the weighted-fusion value $Q$ in multiple directions. The relevant parameters of the substitute values are all trainable.

The network structure of Correct-Edge-FCN can be seen in Figure 6.

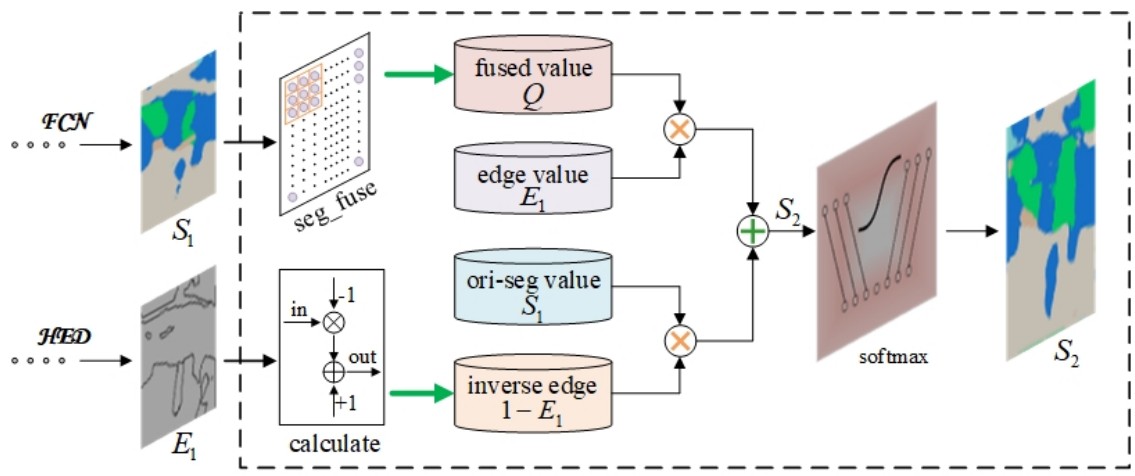

**Figure 6.** Architecture of Correct-Edge-FCN network. This network draws on the Domain Transform algorithm in one-dimensional signals and introduces edge information as a correction coefficient into the semantic segmentation network.

The loss function of Correct-Edge-FCN is basically the same as the loss function of Cascade-Edge-FCN. The difference is that there is no $E_2$. We have:

$$L_1 = 0.5f_1(S_1, GT\_seg) + 0.5f_1(S_2, GT\_seg) \tag{12}$$

$$L_2 = L_{HED} \tag{13}$$

$$L = L_1 + L_2 \tag{14}$$

Correct-Edge-FCN draws on the Domain Transform algorithm in the one-dimensional signal, obtaining the correction value by performing the convolution operation on the original segmentation result. Edge information is to determine the proportion of the original segmentation result and the correction value in the new segmentation result. Compared with Cascade-Edge-FCN, it only adds one more convolution layer, and therefore the computation is relatively simple. As for how to introduce edge information into the segmentation network, we artificially set the rules using the Domain Transform algorithm moreover convolution operation.

## 4. Experiments and Analysis

### 4.1. Dataset

Two datasets are used here: one is the satellite ESAR dataset, and another is optical remote sensing images called GID. ESAR dataset is used for validating our ideas in the field of SAR image semantic segmentation, while GID dataset is for further supporting the ideas in the field of optical remote sensing image semantic segmentation.

There is only one large image in ESAR dataset, with shape of $1187 \times 1187$. Five categories are in the large image (background, arable land, forest, road and building), as is shown in Figure 7.

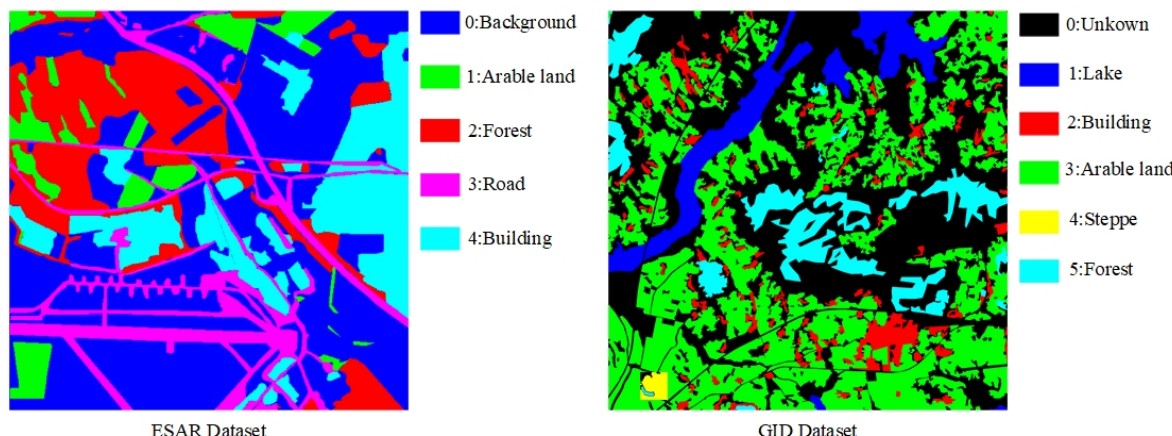

**Figure 7.** Categories of Two Datasets.

To make full use of this large image, we crop it to 400 small images with shape of 256 × 256, which are numbered as 1–400 in order. They are used for 4 times, as is shown in Table 2.

**Table 2.** Procedure for using ESAR.

| Times | Traing Images | Test Images |
|:-----:|:-------------:|:-----------:|
| 1 | 101–400 | 1–100 |
| 2 | 1–100 & 201–400 | 101–200 |
| 3 | 1–200 & 301–400 | 201–300 |
| 4 | 1–300 | 301–400 |

GID dataset consists of 150 large images with size of 6800 × 7200, which are numbered as 1–150. Six categories are in the whole dataset (unknown, lake, building, arable land, steppe and forest), seen also in Figure 7.

We use the 1–80 images for training, 81–96 images for validating, and 97–150 images for testing. For the convenience of training and validating, all the original large images are cropped to 17 × 18 small images with shape of 400 × 400, discarding the image in which unknown part accounts for more than 25%. Finally, 12,292 small images are acquired for training and 1608 small images are acquired for validating. For the test dataset, each image is also cropped to 17 × 18 small images with shape of 400 × 400 before it is fed into the network and spliced to the large image after segmentation. The process of cropping and splicing are all integrated into the whole inference operation, making the network can segment large image end-to-end.

### 4.2. Experiment Setup

In the experiment, the batch size is set as 2. For ESAR dataset, 300 images are used in one training epoch. Therefore, one epoch iterates 150 times. The total epoch is set as 100, equaling to 15,000 iterations. For GID dataset, 12,292 images are trained in one epoch. Therefore, one epoch iterates 6146 times. The total epoch is set as 20, equaling to 122,920 iterations. The learning rate is initially set as $10^{-4}$, whose updated rule is ReduceLROnPlateau in pytorch. The weighting parameters are updated using the Adam optimization algorithm in which $\beta_1$ and $\beta_2$ is set as 0.95 and 0.999.

### 4.3. Evaluation Metrics

The main evaluation method of the experiment is from common semantic segmentation evaluations that are variations on pixel accuracy and region intersection over [4]. They are pixel accuracy *acc*, mean accuracy *mean_acc*, mean intersection over union *mean_iu* and frequency weighted intersection over union *fw_iu*, defined as follows:

$$acc = \frac{\sum_{i=1}^{c} n_{ii}}{\sum_{i=1}^{c} t_i} \tag{15}$$

$$mean\_acc = \frac{1}{c} \sum_{i=1}^{c} \frac{n_{ii}}{t_i} \tag{16}$$

$$mean\_iu = \frac{1}{c} \sum_{i=1}^{c} \frac{n_{ii}}{t_i + \sum_{j=1}^{c} n_{ji} - n_{ii}} \tag{17}$$

$$fw\_iu = \frac{\sum_{i=1}^{c} \frac{t_i n_{ii}}{t_i + \sum_{j=1}^{c} n_{ji} - n_{ii}}}{\sum_{k=1}^{c} t_k} \tag{18}$$

Here, $n_{ij}$ is the number of pixels of class $i$ predicted to belong to class $j$. There are total $c$ different classes, and $t_i = \sum_{j=1}^{c} n_{ij}$ is the total number of pixels of class $i$.

### 4.4. Results

#### 4.4.1. ESAR Results

As stated in Section 4.1, ESAR dataset is used for 4 times, seen in Table 2. That is to say, we have done four training and testing experiments on ESAR dataset. The results of the four sub-experiments are shown in Figure 8. Obviously, the results of these four sub-experiments have proved the superiority of our proposed algorithms. Although the results of the first sub-experiment are not so satisfactory, it does not diminish the value of our algorithms.

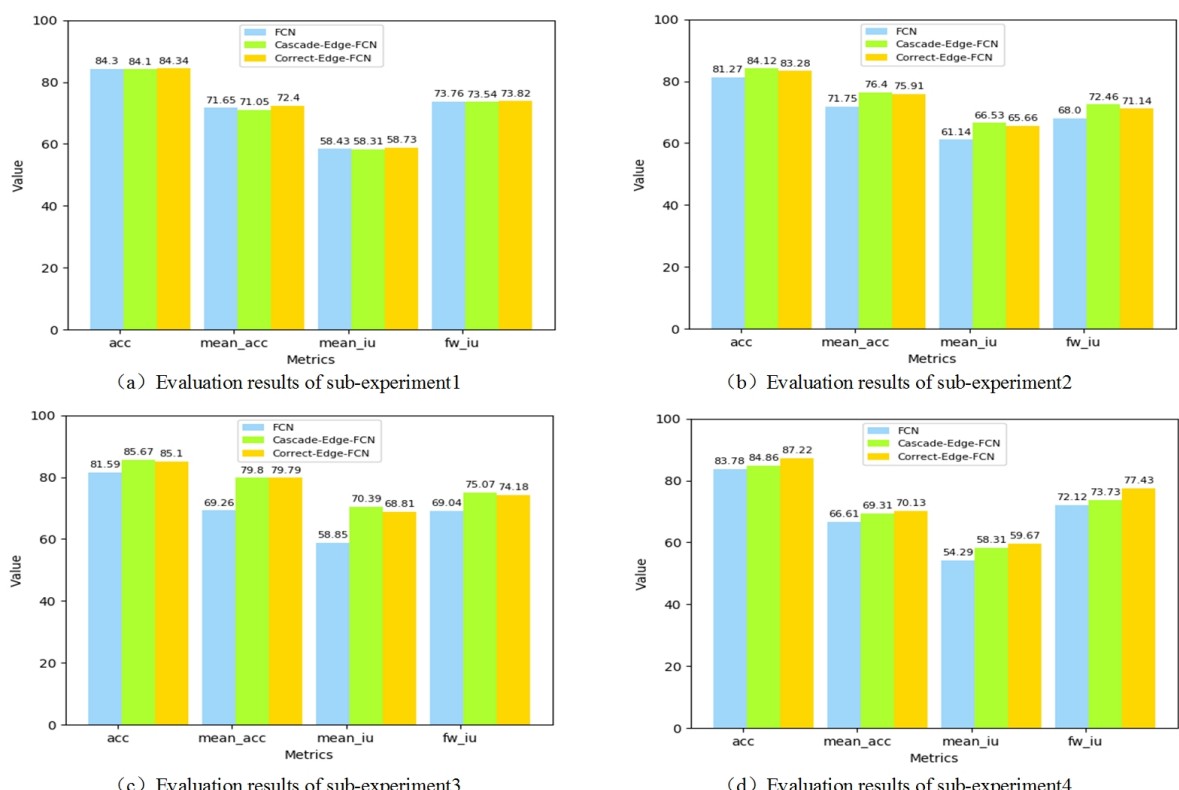

**Figure 8.** Evaluation results of four sub-experiments on ESAR dataset. It can be clearly seen from the height of the "bar" that our proposed algorithm performs better than FCN.

Due to 4 sub-experiments, all 400 small images in the ESAR dataset can be tested. The following discussion is all aimed at the whole dataset.The evaluation results of the 400 small images tested by FCN, Cascade-Edge-FCN, and Correct-Edge-FCN can be seen in Table 3.

**Table 3.** Evaluation Results on ESAR. This table intuitively reflects the segmentation performance on ESAR dataset of the three algorithms as a whole.

| Method | *acc* | *mean_acc* | *mean_iu* | *fw_iu* |
|---|---|---|---|---|
| FCN | 82.74 | 72.22 | 61.87 | 70.47 |
| Cascade-Edge-FCN | 84.69 | 75.88 | 65.76 | 73.54 |
| Correct-Edge-FCN | 84.98 | 76.50 | 66.39 | 73.95 |

It can be seen that Cascade-Edge-FCN and Correct-Edge-FCN both behave better than FCN, and the overall accuracy is increased by 1.9% and 2.1% respectively. The metric *acc* reflects the proportion of all pixels in the entire dataset that are correctly classified, so it is the most intuitive. The *mean_iu* is 3.9% and 4.5% higher than that of FCN8s. Comparing Cascade-Edge-FCN and Correct-Edge-FCN, the overall accuracy of FCN-HED-joint2 is not much different from that of Cascade-Edge-FCN, but *mean_iu* is improved by nearly 0.6%. Difference between these metrics indicates that the Correct-Edge-FCN can better improve category imbalance to a certain extent than Cascade-Edge-FCN, and thus its performance is slightly better than Cascade-Edge-FCN's as a whole.

In the field of machine learning, ROC curve [40] and P-R curve [41] are commonly used to evaluate the performance of classification algorithms. Considering that the number of pixels in the ESAR dataset is still within the manageable range, ROC and P-R curve are given in Figure 9. The ROC and P-R curves of Cascade-Edge-FCN and Correct-Edge-FCN almost overlap, but they are all above the FCN's. The ROC curves' AUC (area under curve) values are all high, but the two Edge-FCNs' AUC values are still 0.5% higher than FCN's. For AUC values of the P-R curves in Figure 9, Cascade-Edge-FCN and Correct-Edge-FCN are respectively 1.9% and 1.6% higher than FCN's. The above comparison once again proves the better performance of the proposed Edge-FCNs.

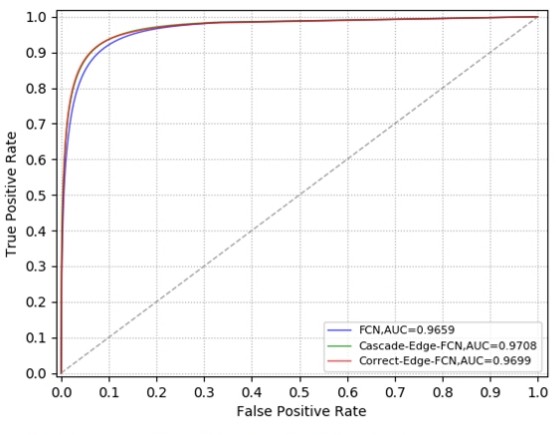 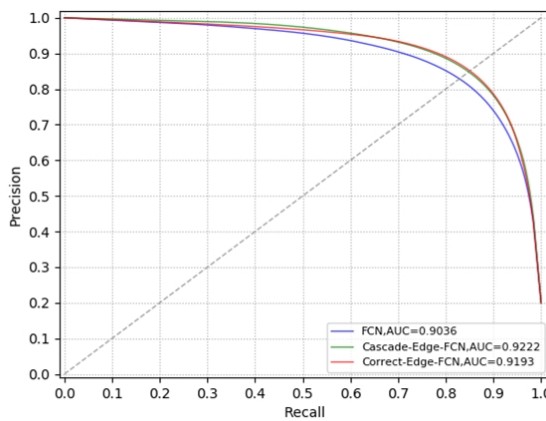

(a).ROC curve of the three algorithms on ESAR dataset　　　　(b).P-R curve of the three algorithms on ESAR dataset

**Figure 9.** ROC and P-R curves of three algorithms on ESAR dataset.

In addition to evaluating the performance of the proposed algorithms as a whole in Table 3, we also give more specific evaluation results. Statistical analysis in Table 4 gives the segmentation accuracy of each landscape category and calculates the mean and standard deviation. The mean value further proves the superiority of the proposed algorithms compared with FCN. Standard deviation reveals imbalance in segmentation between different classes [15]. It can be seen that the standard deviations of these three algorithms are all relatively high due to small amount of image data, even if the proposed algorithm does not improve this problem well.

**Table 4.** Statistical validation on ESAR dataset. "0–4" represents the accuracy of the corresponding category. "Mean" and "std" reflect the statistical characteristics of category accuracy.

| Method | 0 | 1 | 2 | 3 | 4 | *mean* | *std* |
|---|---|---|---|---|---|---|---|
| FCN | 91.44 | 33.80 | 86.01 | 62.27 | 87.56 | 72.22 | 21.77 |
| Cascade-Edge-FCN | 90.67 | 40.37 | 87.94 | 69.69 | 90.71 | 75.88 | 19.41 |
| Correct-Edge-FCN | 90.99 | 42.19 | 86.52 | 74.33 | 88.48 | 76.50 | 18.09 |

Figure 10 shows the proportion of each category in the ESAR dataset and the confusion matrices [42] of the three algorithms. The confusion matrices provide the probability that each class $i$ is divided into category $j$, so the amount of evaluation information is more graceful. To conveniently find the impact of category ratio on classification accuracy, the ratio of each category in ESAR dataset is also given in Figure 9. For categories with relatively few samples such as "1" and "3", the proposed algorithms improve the effect more significantly than FCN. This result may have a precious research value for improving the semantic segmentation of SAR images, because SAR dataset is generally small.

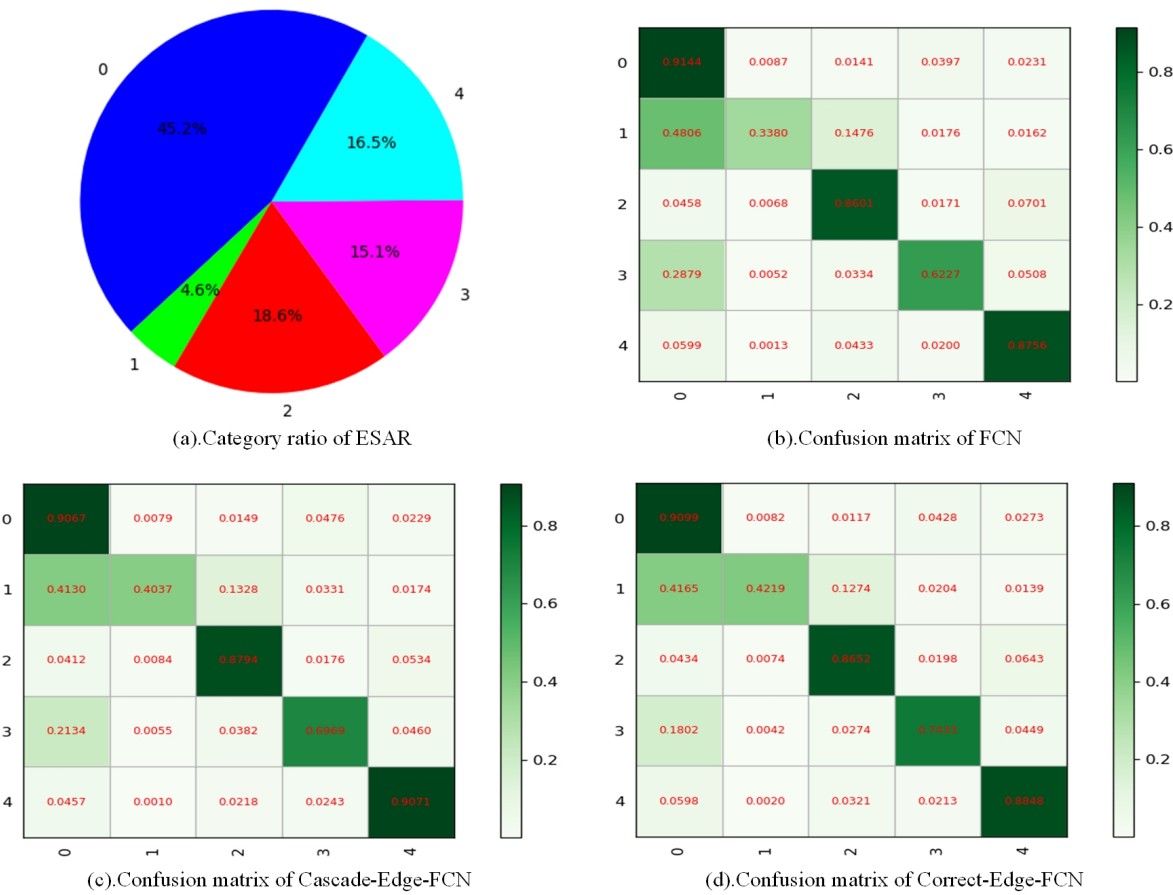

**Figure 10.** Category ratio of ESAR dataset and confusion matrices of three algorithms on ESAR Dataset. It reflects the probability of class $i$ predicted to belong to class $j$.

To see the effect of the edge part, we separately evaluate the edge part. As it can be seen from Table 5, the effects of Cascade-Edge-FCN and Correct-Edge-FCN are still stronger than that of FCN8s. Although the lifting effect is not as large as the whole image, the pixels in the edge part of the image only account for about 2% of the pixel of the whole image. Therefore, it is reasonable that the lifting effect is relatively small.

**Table 5.** Evaluation Results Of edge part on ESAR. Segmentation performance of edge part is not very ideal, but the proposed algorithms can improve it.

| Method | *acc* | *mean_acc* | *mean_iu* | *fw_iu* |
|---|---|---|---|---|
| FCN | 48.02 | 43.02 | 27.91 | 31.10 |
| Cascade-Edge-FCN | 49.38 | 45.45 | 29.88 | 32.51 |
| Correct-Edge-FCN | 50.32 | 45.94 | 30.54 | 33.40 |

When 400 small images are spliced into a large image, the evaluation results are shown in Table 6. Since there are many overlapping parts, there will be lots of pixel points lost during splicing, so the evaluation result will have a certain gap compared with the original result. However, it can be seen that proposed Edge-FCN are still much better than FCN, and Correct-Edge-FCN method is better than Cascade-Edge-FCN.

**Table 6.** Evaluation Results Of large image on ESAR.

| Method | *acc* | *mean_acc* | *mean_iu* | *fw_iu* |
|---|---|---|---|---|
| FCN | 79.09 | 68.39 | 56.49 | 65.63 |
| Cascade-Edge-FCN | 80.82 | 71.58 | 59.92 | 68.27 |
| Correct-Edge-FCN | 80.21 | 71.76 | 59.54 | 67.34 |

The large image of ESAR dataset and its segmentation results of the three algorithms is shown in Figure 11.

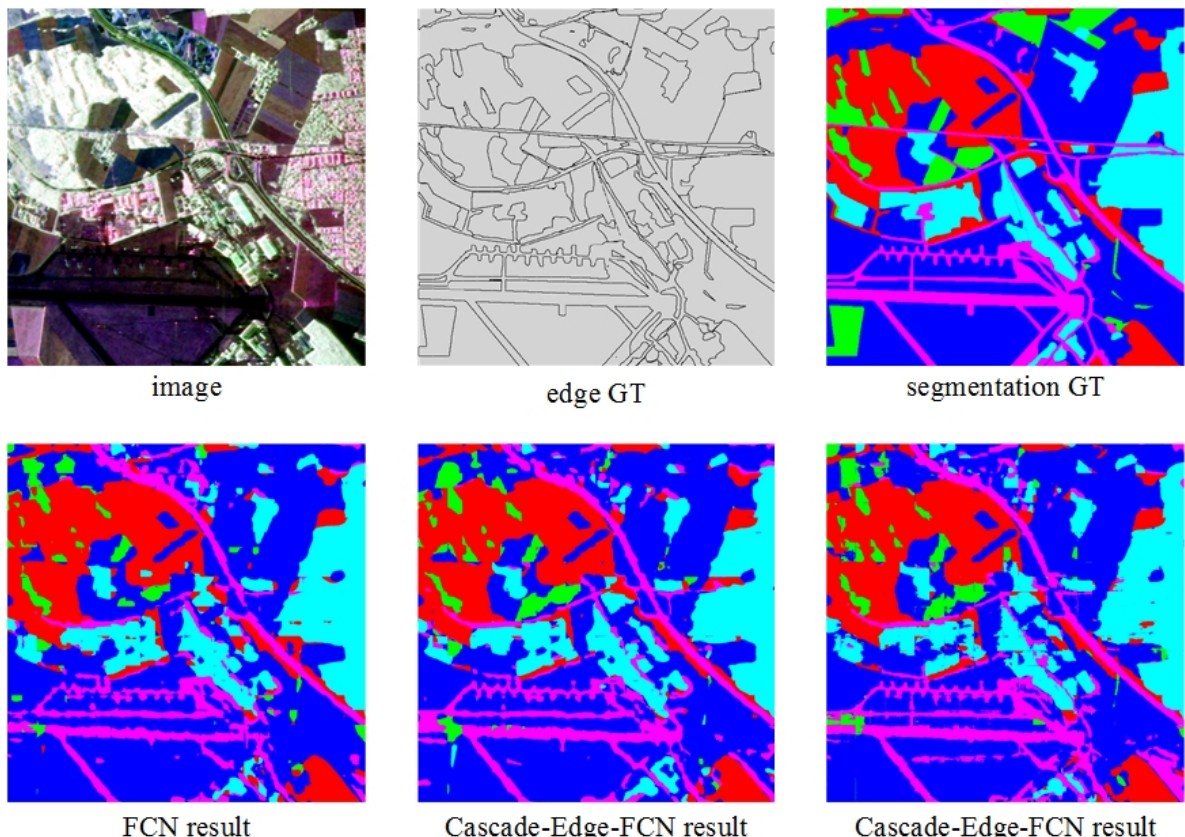

**Figure 11.** Image segmentation results of ESAR Dataset.

4.4.2. GID Results

For the GID dataset, since the label "0" is unknown, we do not evaluate the unknown part. Figure 12 shows each image's segmentation evaluation results. It can be seen that the curves of the algorithm proposed in this paper are generally located above the FCN curve.

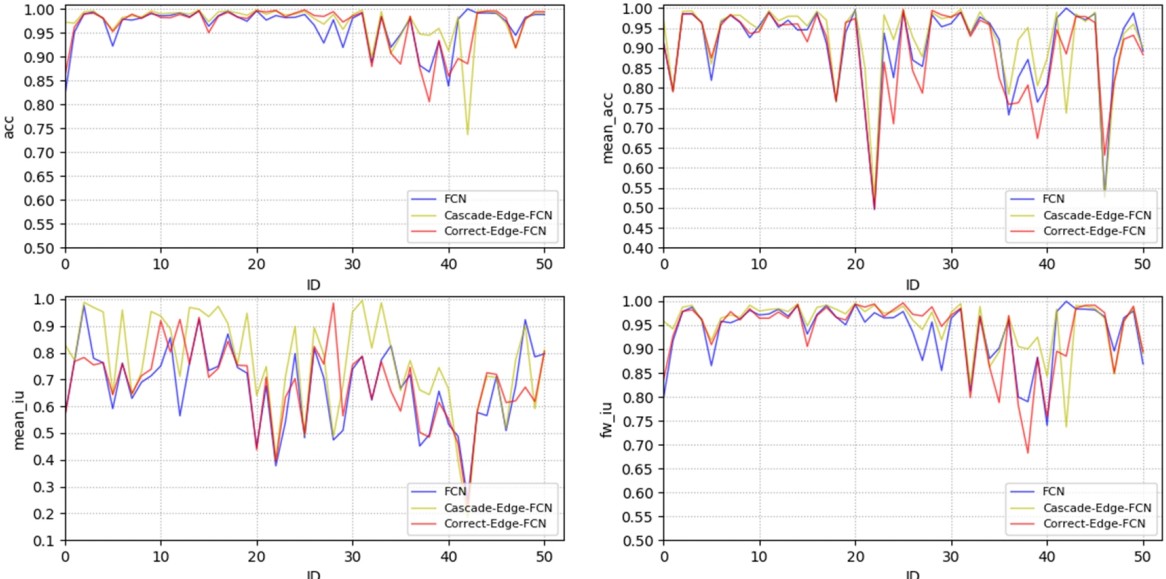

**Figure 12.** Evaluation results of each image in test set of GID. Here, we abandon the images with *acc* < 0.7.

The evaluation result of whole dataset can be seen in Table 7. Evidently, two proposed algorithms introducing edge information both behave better than FCN. Compared with the results on ESAR dataset in Table 3, these four metrics are all higher due to the abundance of training image data.

**Table 7.** Evaluation Results Of three methods on GID dataset. This table intuitively reflects the segmentation performance on GID dataset of the three algorithms as a whole.

| Method | *acc* | *mean_acc* | *mean_iu* | *fw_iu* |
|---|---|---|---|---|
| FCN | 92.73 | 89.30 | 79.57 | 86.88 |
| Cascade-Edge-FCN | 93.90 | 89.99 | 81.06 | 89.06 |
| Correct-Edge-FCN | 94.13 | 88.36 | 80.43 | 89.42 |

Statistical analysis on GID dataset can be seen in Table 8. Compared with the result on ESAR in Table 4, category imbalance in segmentation between different classes is less serious due to GID's large data volume.

**Table 8.** Statistical validation on GID dataset. Here, "1–5" represents the accuracy of the corresponding category. "mean" and "std" reflect the statistical characteristics of category accuracy.

| Method | 1 | 2 | 3 | 4 | 5 | *mean* | *std* |
|---|---|---|---|---|---|---|---|
| FCN | 99.18 | 84.76 | 94.67 | 90.81 | 77.07 | 89.30 | 7.73 |
| Cascade-Edge-FCN | 98.59 | 88.1 | 96.2 | 96.24 | 70.82 | 89.99 | 10.22 |
| Correct-Edge-FCN | 98.48 | 87.05 | 97.54 | 91.29 | 67.43 | 88.36 | 11.27 |

According to Tables 4 and 8, boxplots can be acquired in Figure 12. Obviously, the "boxes" of Edge-FCNs in Figure 13 are above the FCN's, proving the effectiveness of the algorithms in this paper.

The height of the "boxes" in Figure 13, (b) is obviously smaller than that in (a), which also reflects the improvement of the category imbalance due to the larger amount of data.

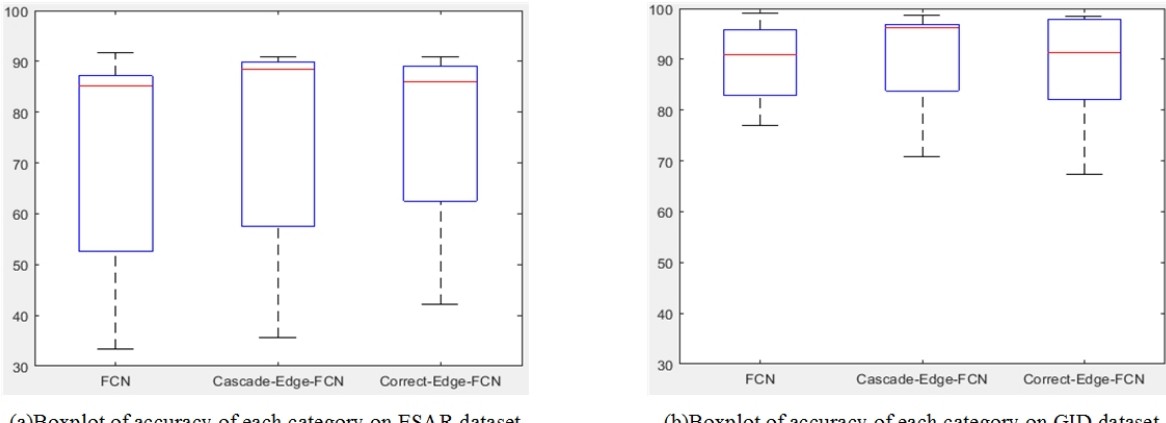

(a)Boxplot of accuracy of each category on ESAR dataset 　　　 (b)Boxplot of accuracy of each category on GID dataset

**Figure 13.** Boxplots of ESAR dataset and GID dataset.

Category ratio of GID dataset and confusion matrices are also given in Figure 14. It can be seen that the segmentation accuracy of each category is all high, which can be evidently noticed by the depth of the matrix color. In contrast, the segmentation accuracy of category "1" in ESAR dataset (seen in Figure 10) is very low. This further proves that large amount of data may improve the classification imbalance problem.

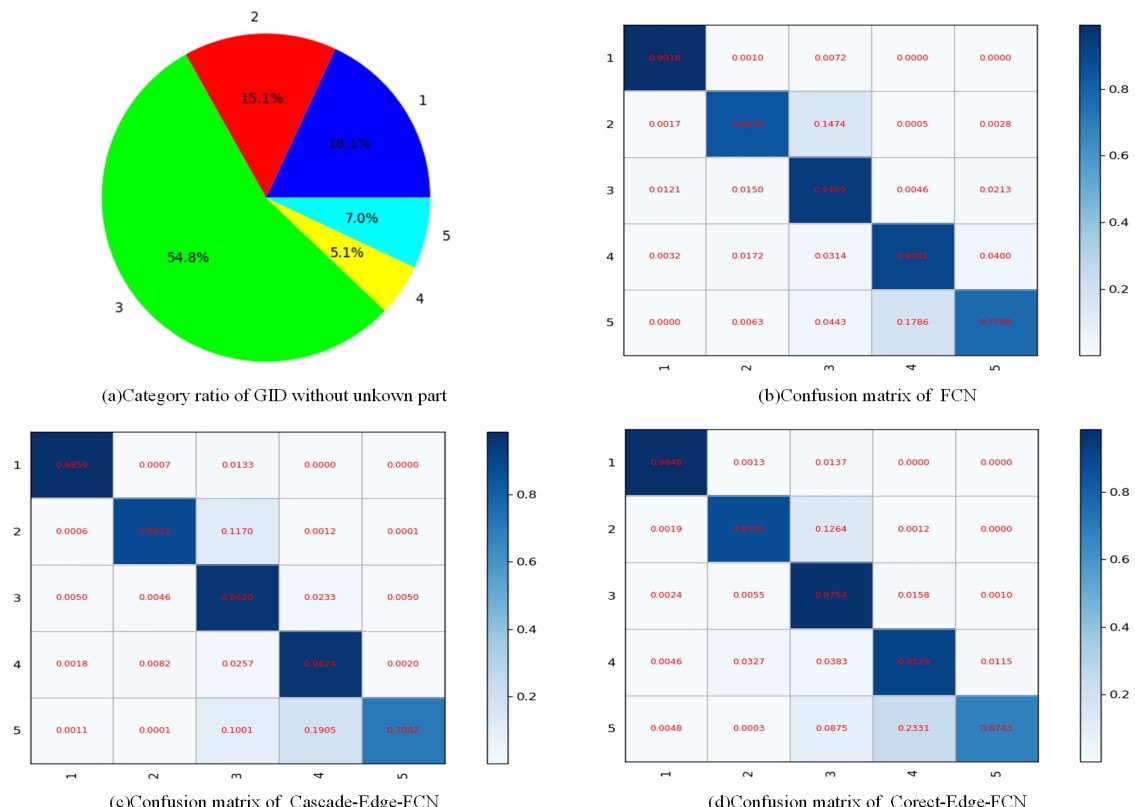

(a)Category ratio of GID without unkown part 　　　　　 (b)Confusion matrix of FCN

(c)Confusion matrix of Cascade-Edge-FCN 　　　　　 (d)Confusion matrix of Corect-Edge-FCN

**Figure 14.** Category ratio of GID dataset and confusion matrices of three algorithms on GID Dataset.

It is obvious that GID is much richer than ESAR in terms of data volume. In the field of deep learning, as the amount of data increases, more potential of the algorithm can be discovered. In other words, the algorithm is more robust. Therefore, the result in Table 7 further verifies the validity of

the algorithm proposed in this paper. Some segmented images is shown in Figure 15. It can be seen evidently that Edge-FCN performs better than FCN.

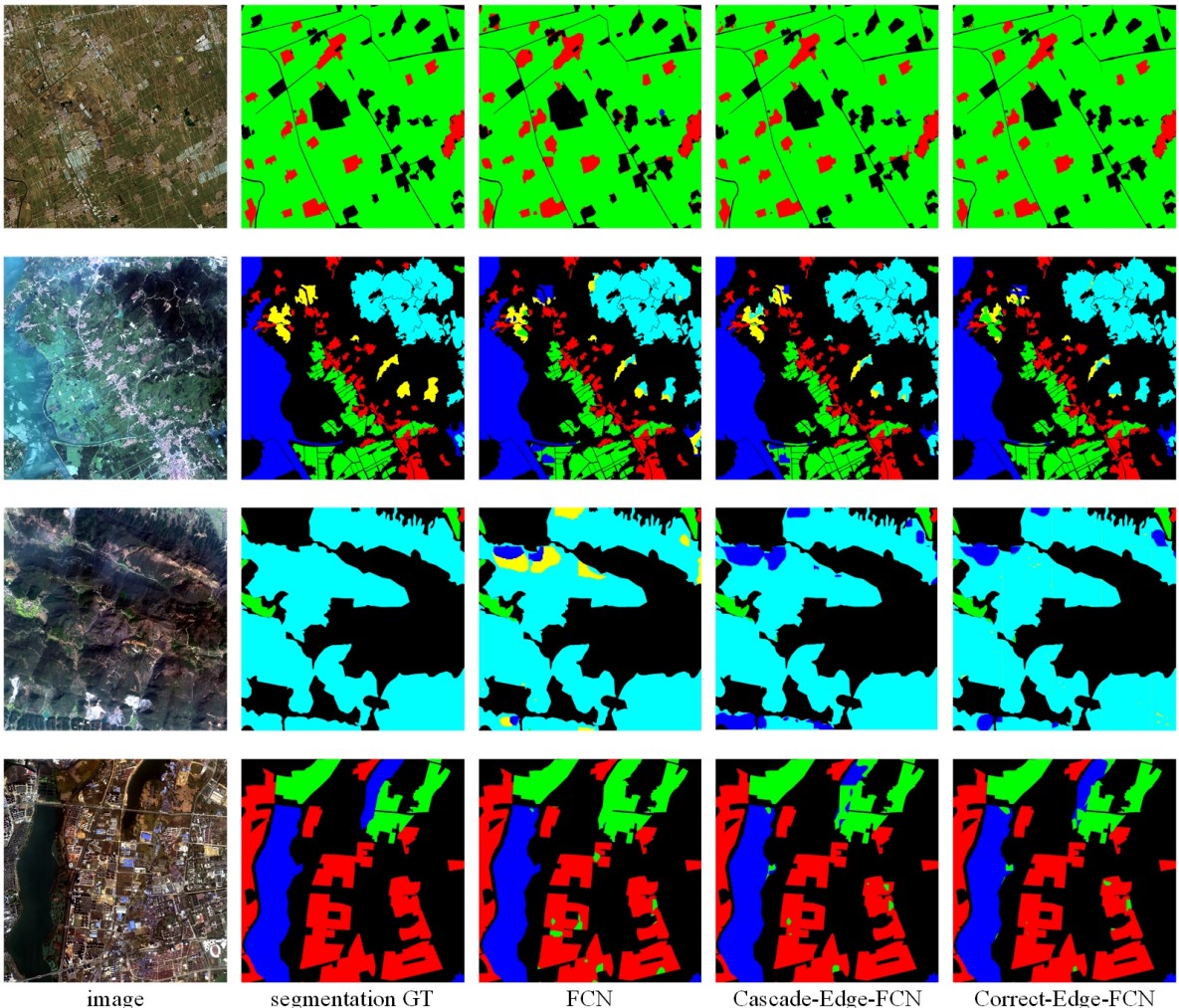

**Figure 15.** Some image samples and the segmentation results of GID Dataset.

### 4.5. Discussion

In Section 4.4, we first evaluate the performance of the algorithms using four commonly used metrics in the field of image semantic segmentation such as *acc*. The results of two datasets both prove the effectiveness of the proposed algorithms compared to FCN. The comparison of ROC and P-R curves on the ESAR dataset further proves this conclusion. After that, statistical analyses are performed on the segmentation accuracy of each class, from which we want to get the improvement for class-imbalance problem of the proposed algorithms. In this regard, the algorithms proposed in this paper have no obvious advantages. However, the comparison between the results of the two datasets shows that increasing the amount of image data can significantly improve this defect. The confusion matrices give the probability that each category can be correctly classified and the probability that it is divided into other wrong categories. Although the information given by the confusion matrices cannot directly reflect the performance of the algorithm, they provide richer information, which is complementary to four intuitive and global metrics such as *acc*.

In summary, although FCN is one of the most excellent networks in image semantic segmentation, its performance still can be improved a lot when edge information is introduced. It can be seen from the experiment that the segmentation result of Edge-FCN always performs better than that of FCN. The main reason is that edge distribution is used as a priori information to guide semantic segmentation,

and the way to guide is done by convolution operation whose parameters can be trained. Furthermore, multiple loss fusion used in the Edge-FCN can deeply supervise the network when training. Compared to Cascade-Edge-FCN, Correct-Edge-FCN introduces edge information and introduces the fusion value of the original segmentation result, so it has better segmentation performance.

## 5. Conclusions

This paper explores the important role of edge information in remote sensing image semantic segmentation. Therefore, a network Edge-FCN that introduces edge information into semantic segmentation is proposed in this paper. Specifically, it is divided into Cascade-Edge-FCN and Correct-Edge-FCN according to the way of introducing edge information. From the experiment results above, it can be obviously known that Edge-FCN can improve the segmentation results compared to directly using FCN. The disadvantage of this algorithm is that a network used for edge detection is added to the original segmentation network, which leads to an increase in the amount of computation. Therefore, for the future work, we intend to realize sharing part of the weights of the segmentation network and edge detection network to reduce the computation.

**Author Contributions:** C.H. proposed the idea and designed the experiments; S.L. performed the experiments and analyzed the results; S.L. wrote the paper; D.X., P.F. and M.L. revised the paper. All authors have read and agreed to the published version of the manuscript.

**Funding:** This work was supported by the National Key Research and Development Program of China (No.2016YFC0803000), and the National Natural Science Foundation of China (No. 41371342 and No. 61331016).

**Conflicts of Interest:** The authors declare no conflict of interest.

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
