# Peer review of "Remote Sensing Image Semantic Segmentation Based on Edge Information Guidance"

_remotesensing, doi:10.3390/rs12091501_

Round 1
Reviewer 1 Report
I am glad to say that I believe authors have taken good care of my previously raised issues: they have been very clear to what they have changed, modified and improved, they have completed my requests and suggestions in a proper way, and manuscript content and quality has been improved now, thus I am suggesting acceptance in its present form.
Reviewer 2 Report
Thank you for incorporating the comments and making the requested corrections.
I would recommend accepting this manuscript for publishing.
This manuscript is a resubmission of an earlier submission. The following is a list of the peer review reports and author responses from that submission.
Round 1
Reviewer 1 Report
This paper presents a novel approach for remote sensing semantic image segmentation and classification based on edge information. Despite the manuscript is well-written and justified, a number of issues have been identified as described next:
MAJOR concerns:
1) In addition to confusion matrices shown in results, I encourage authors to include ROC curve analysis and AUC value computation when evaluating the performance of the proposed remote sensing image classification system:
https://towardsdatascience.com/understanding-auc-roc-curve-68b2303cc9c5
https://www.sciencedirect.com/science/article/pii/S016786550500303X
Also, precision-recall (pr) plots and pr-AUC would be interesting to have a look and compare, and thus are also encouraged:
https://minds.wisconsin.edu/bitstream/handle/1793/60482/TR1551.pdf?sequence=1&isAllowed=y
https://machinelearningmastery.com/roc-curves-and-precision-recall-curves-for-classification-in-python/
2) Proper statistical validation of numerical results shown, including mean +/- standard dev. values or even dispersion boxplots, is encouraged:
Minor comments:
a) Please, include a proper valid reference citation for softmax non-linearity, like for instance:
http://citeseerx.ist.psu.edu/viewdoc/download?doi=10.1.1.310.3078&rep=rep1&type=pdf
b) In general, figure and table captions are quite limited. Please provide further details in order to improve manuscript readability and avoid potential readers to go and seek thought the entire manuscript text for relevant information concerning figures/tables.
Author Response
Manuscript ID: remotesensing-703809
Type: Article
Title: "Remote Sensing Image Semantic Segmentation Based on Edge Information Guidance"
Dear Dr. Fay Riordan:
Thank you very much for your attention and comments on our paper "Remote Sensing Image Semantic Segmentation Based on Edge Information Guidance". Those comments are all valuable and very helpful for revising and improving our paper, as well as the important guiding significance to our researches. We have studied comments carefully and have made correction which we hope meet with approval. Revised portion are marked in yellow in the paper. The main corrections in the paper and the responds to the reviewer’s comments are as flowing:
Point 1: In addition to confusion matrices shown in results, I encourage authors to include ROC curve analysis and AUC value computation when evaluating the performance of the proposed remote sensing image classification system.Also, precision-recall (pr) plots and pr-AUC would be interesting to have a look and compare, and thus are also encouraged.
Response 1:Drawing the ROC curve and precision-recall curve requires the probability values of each pixel in the image to be classified into various categories, but we only retain the final classification results.We had intended to go back to Wuhan University and use the trained model to get the required probability value,but due to the spread of 2019-nCov(novel coronavirusSARI) in Wuhan,we can’t go back to school before the deadline and we can’t determine when we can be allowed to return to school.Sincerely apologize to you and hope you can forgive us.Although we can't give you the ROC curve and PR curve you want to see, we have done more analysis on the experiment.
Point 2: Proper statistical validation of numerical results shown, including mean +/- standard dev. values or even dispersion boxplots, is encouraged.
Response 2:Sincerely appreciate your suggestions.We have added a statistical analysis of the accuracy of each category in the two datasets,as shown in Tables 4 and Table 8 in the revised manuscript.Relatedt analysis can also be found in the text of marked yellow.
Point3:Please, include a proper valid reference citation for softmax non-linearity.
Response 3:Thank you for reminding us of this point.We have added references citation for softmax non-linearity on page 4,line 119.
Point4:In general, figure and table captions are quite limited. Please provide further details in order to improve manuscript readability and avoid potential readers to go and seek thought the entire manuscript text for relevant information concerning figures/tables.
Response 4:We are very sorry for our negligence of adding enough figure and table captions.We have added more captions in the revised manuscript.
Special thanks to you for your good comments!Best regards!

Reviewer 2 Report
In this paper, the authors proposed the Edge-FCN method of semantic segmentation for remote sensing image. Edge-FCN uses the edge information detected by Holistically-Nested Edge Detection(HED) network to correct the FCN’s segmentation results. The topic is interesting, and the paper is organized well.
The results of the compared segmentation methods are very similar. One may wonder whether this improvement of results with the use of Correct Edge-FCN is worth increasing the computing power, although the authors also in their conclusions drew attention to this problem.
The confusion matrices in fig. 8 and fig.10 are not described in the text. It is difficult to notice the correlation between these and tables 3 and 6. In addition, fig. 8 and 10 are not indistinct.
Description to fig. 11 does not explain what joint1 and join2 mean.
The problem of semantic analysis of image content is also solved by texture analysis, among others with the use of CNN. I suggest authors could review and add these papers in the reference, for example:
[1] “A comparison of texture measures for the per-field classification of mediterranean land cover”, International Journal of Remote Sensing, vol. 25, no. 19, pp. 3943–3965, 2004.
[2] “Road Traffic Conditions Classification Based on Multilevel Filtering of Image Content Using Convolutional Neural Networks”. 2018. IEEE Intell. Transp. Syst. Mag. 2018. Vol. 10 (3), p. 11–21.
[3] “Convolutional neural networks for large-scale remote-sensing image classification”, IEEE Trans. Geosci. Remote Sens., vol. 55, no. 2, pp. 645–657, 2017.
In conclusion, I say that the article is interesting and valuable. Minor errors do not diminish its scientific value.
Author Response
Manuscript ID: remotesensing-703809
Type: Article
Title: "Remote Sensing Image Semantic Segmentation Based on Edge Information Guidance"
Dear Reviewer:
Thank you very much for your attention and comments on our paper "Remote Sensing Image Semantic Segmentation Based on Edge Information Guidance". Those comments are all valuable and very helpful for revising and improving our paper, as well as the important guiding significance to our researches. We have studied comments carefully and have made correction which we hope meet with approval. Revised portion are marked in yellow in the paper. The main corrections in the paper and the responds to the reviewer’s comments are as flowing:
Point 1:The results of the compared segmentation methods are very similar. One may wonder whether this improvement of results with the use of Correct Edge-FCN is worth increasing the computing power, although the authors also in their conclusions drew attention to this problem.
Response 1:Sincerely thank you for the comments on this issue.What we want to do is to explore the impact of edge information on the semantic segmentation of remote sensing images, so the computation cost is not considered temporarily.The experiment results prove the validity of our idea,which we think is the most important contribution of this paper.As for the amount of computation, we will study it in future work.Thanks again for your comment.
Point 2: The confusion matrices in fig. 8 and fig.10 are not described in the text. It is difficult to notice the correlation between these and tables 3 and 6. In addition, fig. 8 and 10 are not indistinct.
Response 2:We have added the description of fig.8 and fig.10 and distinguished them with different colors.The correlation between table 3 and 6 have been described on Page 15 in the revised manuscript.Note that because we have revised the manuscript, fig.8 and fig.10 have become fig.9 and fig.11, and table 6 has become table 7.Thank you for your useful advice,which can improve our manuscript readability.
Point 3:Description to fig. 11 does not explain what joint1 and join2 mean.
Response 3:”joint1” and “joint2” are the abbreviations of the two algorithms proposed in the paper at the beginning. We are sorry for not correcting it in time.Now it has been corrected it in fig.11.Thank you for pointing out our negligence.
Point 4:The problem of semantic analysis of image content is also solved by texture analysis, among others with the use of CNN. I suggest authors could review and add these papers in the reference, for example:
[1] “A comparison of texture measures for the per-field classification of mediterranean land cover”, International Journal of Remote Sensing, vol. 25, no. 19, pp. 3943–3965, 2004.
[2] “Road Traffic Conditions Classification Based on Multilevel Filtering of Image Content Using Convolutional Neural Networks”. 2018. IEEE Intell. Transp. Syst. Mag. 2018. Vol. 10 (3), p. 11–21.
[3] “Convolutional neural networks for large-scale remote-sensing image classification”, IEEE Trans. Geosci. Remote Sens., vol. 55, no. 2, pp. 645–657, 2017.
Response 4:Thank you for your advice.We have reviewed the papers and added them in the reference.It can be seen on page 2 maked yellow.
Special thanks to you for your good comments!Best regards!

Round 2
Reviewer 1 Report
Unfortunately, authors have only taken my concerns in part, not responding or completing properly various of my raised concerns, as listed next in A) , B) and C) new Major concern and comments points:
A) Point 1: In addition to confusion matrices shown in results, I encourage authors to include ROC curve analysis and AUC value computation when evaluating the performance of the proposed remote sensing image classification system.Also, precision-recall (pr) plots and pr-AUC would be interesting to have a look and compare, and thus are also encouraged.
Response 1:Drawing the ROC curve and precision-recall curve requires the probability values of each pixel in the image to be classified into various categories, but we only retain the final classification results.We had intended to go back to Wuhan University and use the trained model to get the required probability value,but due to the spread of 2019-nCov(novel coronavirusSARI) in Wuhan,we can’t go back to school before the deadline and we can’t determine when we can be allowed to return to school.Sincerely apologize to you and hope you can forgive us.Although we can't give you the ROC curve and PR curve you want to see, we have done more analysis on the experiment.
I am so sorry about the situation in Wuhan concerning coronavirus, and that you have had not the chance to go to school and complete the requests I raised, but I still believe Major Concern Point 1 is mandatory, even if you need additional time to complete, repeated next in its original format for your convenience:
MAJOR concerns:
1) In addition to confusion matrices shown in results, I encourage authors to include ROC curve analysis and AUC value computation when evaluating the performance of the proposed remote sensing image classification system:
https://towardsdatascience.com/understanding-auc-roc-curve-68b2303cc9c5
https://www.sciencedirect.com/science/article/pii/S016786550500303X
Also, precision-recall (pr) plots and pr-AUC would be interesting to have a look and compare, and thus are also encouraged:
https://minds.wisconsin.edu/bitstream/handle/1793/60482/TR1551.pdf?sequence=1&isAllowed=y
https://machinelearningmastery.com/roc-curves-and-precision-recall-curves-for-classification-in-python/
B) Point 2: Proper statistical validation of numerical results shown, including mean +/- standard dev. values or even dispersion boxplots, is encouraged.
Response 2:Sincerely appreciate your suggestions.We have added a statistical analysis of the accuracy of each category in the two datasets,as shown in Tables 4 and Table 8 in the revised manuscript.Relatedt analysis can also be found in the text of marked yellow.
Authors have included some std. values but no boxplots have been included yet. Again encourage authors to make use of dispersion plots like boxplots.
C) Minor comments:
a) Please, include a proper valid reference citation for softmax non-linearity, like for instance:
http://citeseerx.ist.psu.edu/viewdoc/download?doi=10.1.1.310.3078&rep=rep1&type=pdf
Point 3:Please, include a proper valid reference citation for softmax non-linearity.
Response 3:Thank you for reminding us of this point.We have added references citation for softmax non-linearity on page 4,line 119.
Unfortunately, included recent (2018) reference while being related to softmax non-linearity function use but it should be more fair to include original use citation about softmax non-linearity, like the one suggested (1999), which was published almost 20 years before the one included by authors:
http://citeseerx.ist.psu.edu/viewdoc/download?doi=10.1.1.310.3078&rep=rep1&type=pdf